# Target Mechanisms of the Cyanotoxin Cylindrospermopsin in Immortalized Human Airway Epithelial Cells

**DOI:** 10.3390/toxins14110785

**Published:** 2022-11-11

**Authors:** Sabine Ziesemer, Susann Meyer, Julia Edelmann, Janita Vennmann, Celine Gudra, Denise Arndt, Marcus Effenberg, Olla Hayas, Aref Hayas, Johanna Sophia Thomassen, Barbara Kubickova, Dierk-Christoph Pöther, Jan-Peter Hildebrandt

**Affiliations:** 1Animal Physiology and Biochemistry, University of Greifswald, Felix Hausdorff-Strasse 1, D-17489 Greifswald, Germany; 2Federal Institute for Occupational Safety and Occupational Medicine, Nöldnerstrasse 40-42, D-10317 Berlin, Germany; 3RECETOX, Faculty of Science, Masaryk University, Kotlarska 2, CZ-61137 Brno, Czech Republic

**Keywords:** cyanotoxin, cylindrospermopsin, airway epithelial cells, proteomics, cytokinesis, cell cycle, extracellular matrix

## Abstract

Cylindrospermopsin (CYN) is a cyanobacterial toxin that occurs in aquatic environments worldwide. It is known for its delayed effects in animals and humans such as inhibition of protein synthesis or genotoxicity. The molecular targets and the cell physiological mechanisms of CYN, however, are not well studied. As inhalation of CYN-containing aerosols has been identified as a relevant route of CYN uptake, we analyzed the effects of CYN on protein expression in cultures of immortalized human bronchial epithelial cells (16HBE14o^−^) using a proteomic approach. Proteins whose expression levels were affected by CYN belonged to several functional clusters, mainly regulation of protein stability, cellular adhesion and integration in the extracellular matrix, cell proliferation, cell cycle regulation, and completion of cytokinesis. With a few exceptions of upregulated proteins (e.g., ITI inhibitor of serine endopeptidases and mRNA stabilizer PABPC1), CYN mediated the downregulation of many proteins. Among these, centrosomal protein 55 (CEP55) and osteonectin (SPARC) were significantly reduced in their abundance. Results of the detailed semi-quantitative Western blot analyses of SPARC, claudin-6, and CEP55 supported the findings from the proteomic study that epithelial cell adhesion, attenuation of cell proliferation, delayed completion of mitosis, as well as induction of genomic instability are major effects of CYN in eukaryotic cells.

## 1. Introduction

Harmful algal blooms occur in eutrophicated water bodies during summer due to high nutrient content (esp. nitrogen and phosphorus) and intense sunlight [1]. Accumulations of toxin-producing algae or cyanobacteria occur in freshwater lakes, coastal marine areas, or brackish water [2,3]. Cyanobacteria may reach extreme densities under such conditions and produce toxins that may be harmful to animals and humans when ingested by drinking contaminated water or inhaled by inspiring aerosols containing cyanobacteria and/or cyanotoxins [4,5,6,7,8]. Exposure of humans may occur in recreational settings but also occupational settings during the aerosolization of water containing cyanobacteria and their toxins [9,10].

One of these toxins is cylindrospermopsin (CYN) [11,12]. While most CYN-producing cyanobacteria (e.g., *Cylindrospermopsis raciborskii*) occur in tropical or subtropical areas, it is expected that they invade water bodies in temperate regions due to the current climate change [13,14]. CYN-producing cyanobacteria (e.g., *Aphanizomenon flos-aquae* or *Anabaena* sp.) have already been detected in temperate zones [13,15,16,17,18].

CYN is a metabolically stable [19] tricyclic uracil derivative (Figure 1, insert) that is well soluble in water due to its zwitterionic character [20]. The lethal dose in mice (gavage) is 4.4–6.9 mg/kg (corresponds to 11–17µmol/L) [21]. The main target of CYN in vertebrates, including humans, is the liver but other organs are affected as well [22]. CYN may be metabolized and bioactivated by liver enzymes (e.g., cytochrome p450 monooxygenases), which would enhance its toxicity [23], but this hypothesis has been challenged by conflicting experimental results [24], as thoroughly discussed by Raška et al. [25]. CYN has been implicated in the attenuation of protein synthesis and induction of cell death in hepatocytes with effective concentrations in the lower micromolar range [26,27,28,29]. CYN induced production of reactive oxygen species and lipid peroxidation in HepG2 cells [30]. Chronic exposure of liver cells to CYN may compromise DNA integrity by inducing double-strand breaks and cell cycle arrest [31,32]. Attenuation of cell proliferation by exposure to CYN has been observed in several cell systems, CHO-K1 [33], HepG2 [32], and human peripheral lymphocytes [34]. These findings have prompted the WHO to set limits on human CYN exposure (guideline values, GVs) by drinking water to 0.7 (lifetime drinking water), 3 (short-term drinking water), or 6 µg/L (recreational exposure), respectively [35]. More recently, researchers have identified other potential target cells in animals and humans such as intestinal [36,37] or airway epithelial cells [38]. In airway epithelial cells, CYN exposure in the lower micromolar range induces a reduction in cell proliferation rate due to the downregulation of cyclin D1 and an extension of the metaphase of the cell cycle [39].

The molecular targets of CYN and the cell physiological mechanisms in animal or human cells affected by CYN, however, are not well studied yet. We used a proteomic approach to identify proteins whose abundances were affected in CYN-exposed human immortalized airway epithelial cells (16HBE14o^−^ cells). This approach allowed the identification of potential targets of CYN in these cells. Furthermore, we sorted the regulated proteins in functional clusters, which gave hints on the cell physiological alterations that are brought about by CYN exposure of cells. Three of the potential target proteins (Claudin-6, SPARC, and CEP55) were analyzed in detail by Western blotting in extracts of CYN-exposed cells to verify the observations of the proteomic part of the study.

## 2. Results and Discussion

Cylindrospermopsin (CYN) does not have major immediate toxic effects on 16HBE14o^−^ human airway epithelial cells, but it activates Erk-type- and p38MAP-kinases after at least 8 h of CYN exposure which results in attenuations of cell spreading, cell-cell- and cell-matrix adhesion and tightness of the cell layers [38]. With similar delays, cells reduced the rate of cell proliferation, and dividing cells showed extensions of the metaphase during mitosis [39]. In an attempt to identify molecular targets of CYN in these cells that may be causally associated with these cell physiological phenomena, we exposed 16HBE14o^−^ cells to 5 µmol/L CYN for 20 h and analyzed the proteomes of these cells in comparison with those of vehicle-treated control cells.

Approximately 5000 proteins were identified in 16HBE14o^−^ cells (Appendix A), of which 98 showed CYN-mediated alterations in abundance with an adjusted *p*-value ≤ 0.05 in two of three replicates and changes in abundance ratio ≥ 1.5 in two of three replicates. Most of these proteins (76) were downregulated in CYN-exposed cells, while only 14 were upregulated (Appendix A). A potential explanation for this finding is that CYN may affect the synthesis or degradation rates of proteins with short lifespans, a conclusion that confirms previous reports in other cell types [26,28,29]. Figure 1 shows a synopsis of major cell physiological mechanisms that seemed to be affected by the CYN exposure of 16HBE14o^−^ cells ([39], this study). For an initial overview of affected biological processes, UniProtKB-accessions of significantly regulated proteins were subject to functional enrichment analysis by g:GOSt of g:Profiler [40] for GO—biological processes and GO—molecular functions or by overrepresentation analysis in Reactome pathways [41] (Appendix A). Results from g:Profiler indicated several proteins with inhibitory activity on (endo-) peptidases [GO:0004866, GO:0030414,and GO:0004867] (Appendix A). Several types of proteinase inhibitors (Cystatin-C (P01034) and SPINT2 (O43291)) were significantly downregulated (mean ratios of expression levels in CYN-treated versus vehicle-treated cells (ratio CYN/Ctrl): Cystatin 0.253, SPINT-2: 0.123), which may support the conclusion that protein stability may be reduced upon CYN exposure of cells (Appendix A). However, the heavy chains two and three of the inter-α-trypsin inhibitor (P19823 and Q06033), which affects extracellular matrix components [42], were approximately three-fold upregulated (Appendix A), indicating that certain proteins may be selectively protected from degradation in CYN-exposed cells. On the other hand, the synthesis of membrane or secretory proteins may also be affected by CYN, as the ribosome binding protein 1 (Q9P2E9), which assists in coupling ribosomes to the cytosolic surface of the endoplasmic reticulum, was strongly downregulated (ratio CYN/Ctrl: 0.18).

Cell adhesion and extracellular matrix (ECM) proteins mediate the integration of individual cells into functional tissue structures. In particular, epithelial cells form diffusion barriers by building cell layers with a strong attachment of the basolateral plasma membranes to the underlying basal lamina and adhesion junctions for stabilizing cell layers against mechanical stress [43] or tight junctions for controlling the permeability of the paracellular pathway [44]. CYN may be able to reduce the strength of cell-cell- and cell-matrix-adhesion (Appendix A) as indicated by the findings of several proteins involved in these functions that were downregulated in our proteomic analysis (Appendix A). SPARC (secreted protein acidic and rich in cysteine; P09486) [45,46,47], an extracellular calcium-binding protein regulating the docking of epithelial cells to collagen fibers, was strongly downregulated (ratio CYN/Ctrl: 0.143). This result was confirmed by the results of Western blot experiments on whole cell extracts of CYN-exposed 16HBE14o^−^ cells (Figure 2), indicating that CYN may compromise the coherence of epithelial cell layers. Furthermore, the proteomic analyses revealed that other matrix-associated proteins, such as agrin (O00468; [48]) and laminin (subunit α-5; O15230; [49]), were substantially downregulated as well (ratios CYN/Ctrl: 0.23 or 0.387, respectively; Appendix A). This in conjunction with the observation that TIMP2 (P16035; [50]), an inhibitor of ECM-digesting metalloproteinases, was downregulated in CYN-exposed cells (ratio CYN/Ctrl: 0.25) similar to the previously mentioned protease inhibitors SPINT-2 and ITIHs, indicating that CYN-exposure of epithelial cells may severely disturb their matrix adhesion.

Claudins are proteins of the tight junctions which determine the paracellular permeability of epithelial cell layers and transepithelial electrical resistance [51,52]. In the proteomic analyses, the level of claudin-6 (P56747; [53]) seemed to be somewhat diminished in CYN-exposed 16HBE14o^−^ cells, although the change was not significant (ratio CYN/Ctrl: 0.64; Appendix A). Western blot analyses confirmed that this may only be a very small effect that is only visible when expression levels in CYN-exposed cells are compared with those in vehicle-treated cells (Figure 2). There seems to be a positive trend in claudin-6 abundance in vehicle-treated control cells over 36 h that may be due to an increasing number of tight junctions in maturing epithelial cell sheets and, concomitantly, the production of more claudin-6 molecules. CYN at high concentrations (5 µmol/L) and long exposure times (36 h) may indeed compromise this additional portion of claudin-6 production but does not seem to have a negative effect on the claudin-6 molecules that are already in place. Assuming that this interpretation of the data is correct, one can conclude that long-lived proteins may be less affected in their abundance by the CYN exposure of cells than short-lived ones. In any case, CYN exposure of these epithelial cells seems to inhibit the maturation of functional tight junctions, which may negatively affect the selectivity of tight junctions toward ions and small molecules.

Upon completion of DNA replication during the S phase of the cell cycle, eukaryotic cells prepare for mitosis. Formation of the mitotic spindle, a bipolar arrangement of bundles of microtubules, is the first step [54]. Centrosomes organize the formation of the spindle apparatus in mitotic cells and serve as signaling centers that control essential steps of spindle function [55]. Many accessory and adaptor proteins are involved in the organization of the spindle apparatus and the coupling of the centromeric regions of each of the sister chromatids to the spindle microtubules via the kinetochores [56]. The general architecture of mitotic epithelial cells in meta-, ana- and telophases is illustrated in Figure 3 using example images of 16HBE14o^−^ cells. One of the spindle-associated proteins is FAM83D (Q9H4H8), which directs casein kinase 1 to the spindle for proper positioning and reorganization of microtubules during the separation of the two sets of sister chromatids [57]. The downregulation of FAM83D (Q9H4H8) in CYN-treated cells (ratio CYN/Ctrl: 0.35; Appendix A) may be another factor responsible for the observed metaphase extension. Another accessory protein mediating proper spindle orientation is the suppressor anaphase-promoting complex domain 2 (SAPCD2; Q86UD0) [58,59]. The level of this protein dropped substantially, nearly reaching our criteria of significance in CYN-exposed cells (ratio CYN/Ctrl: 0.253; Appendix A), a process that may also contribute to metaphase extension. Last but not least, the centrosomal protein of 55 kD (CEP55; Q53EZ4) is another important accessory protein enabling cells to successfully pass mitosis [53,60]. At the onset of metaphase, this protein is associated with the centrosomes. It is phosphorylated by Erk2 and Cdk1, a spindle assembly checkpoint protein kinase activated by cyclin B [61], and then also located at the midbody [62], where it is required for successful completion of cytokinesis (Figure 3). Fluorescent microscopic inspection of normal 16HBE14o^−^ cells during ana- and telophases (Figure 3C) showed that CEP55 is associated with spindle microtubules as well as with midbody microtubules. Proteomic analyses on protein extracts of cells exposed to CYN indicated that CEP55 is downregulated to approximately 50% of the control level (ratio CYN/Ctrl: 0.523; Appendix A), although this change did not meet the criteria of significance. Semi-quantitative Western blotting results, however, showed strong negative effects of CYN (2.5 or 5 µmol/L for 24 or 36 h) on the abundances of CEP55 in these cells (Figure 2). Downregulation of CEP55 in CYN-exposed cells may have contributed to the observation that such cells were no longer able to exit from mitosis at their normal pace [39].

The effect of CYN on the metaphase duration may be one of the potential explanations for the observation that CYN reduced the overall rate of cell proliferation in 16HBE14o^−^ cell cultures starting after 8 h of CYN exposure [39]. The search for cell cycle-regulatory proteins in the proteome data that may constitute further points of interference gave ambiguous results. Some inhibitors of cyclin/Cdk-complexes, such as p16Ink4a (P42771; [63]) or p21Cip1 (P38936; [64]), were substantially downregulated in CYN-exposed cells (ratios CYN/Ctrl: 0.44 or 0.05, respectively; Appendix A). Such a condition should promote cell proliferation. CYN-mediated downregulation of tumor suppressor Arf (Q8N726) (ratio CYN/Ctrl: 0.22), which may induce cell cycle arrest in G1 or G2 phases of the cell cycle [65,66], should have similar effects. On the other hand, the downregulation of ASF1B (Q9NVP2) (ratio CYN/Ctrl: 0.28; Appendix A), a histone chaperone that mediates nucleosome assembly and disassembly and assists in DNAreplication [67], may attenuate the rate of cell proliferation. The mitotic cyclin B1 (P14635) that controls the G2 checkpoint of the cell cycle and allows its propagation from metaphase to anaphase [68] was detected in the proteomic analysis, but its expression level was not altered by CYN-exposure of cells (Appendix A), a finding that confirmed previous Western blotting results [39]. The G1/S-specific cyclin D1 (P24385), which has been previously identified as a cell cycle regulator that was significantly downregulated in Western blot analyses of CYN-treated 16HBE14o^−^ cells [39], was not identified in the proteomic assays of extracts of CYN-exposed cells. However, cyclin D3, which has similar functions as the D1 isoform, was shown to be strongly downregulated in cells exposed to CYN (ratio CYN/Ctrl: 0.01; Appendix A). As cyclin D is essential for the entry of cells into the cell cycle, we assume that CYN-mediated downregulation of these cyclins may have a limited cell proliferation rate in CYN-treated cells.

Inspection of the 16HBE14o^−^ cells using fluorescence microscopy revealed that CYN at higher concentrations (2.5 and 5 µmol/L) and longer exposure times (24 h and 36 h) induced subtle increases in the proportions of irregularly shaped cell nuclei and micronuclei in the cells when compared with the initial situations at 0 h (Figure 4). This indicates that CYN may induce chromosomal instability and supports the hypothesis of CYN-induced genotoxicity in eukaryotic cells [31,32,34,69,70]. A potential mechanism of how genome instability could be induced by CYN exposure of cells may be derived from the observation in the proteomic investigations that RecQ-mediated genome instability protein 2 (Q96E14) is downregulated by two-thirds of its original level in CYN-exposed cells (ratio CYN/Ctrl: 0.303; Appendix A). Loss of this protein has been previously shown to increase genome instability in human cells [71].

There is, to our knowledge, only one previous proteomic investigation of CYN’s effects on eukaryotic cells [72]. In human hepatoma cells (HepG2) exposed to CYN (2.5 µmol/L for 24 h), the cell proliferation rate was attenuated, and signs of genotoxicity were observed. The proteins that were altered in their abundancies have been implicated in biological processes such as protein folding, antioxidant defense, energy metabolism, cell anabolism, cell signaling, and maintenance of the cytoskeleton. Although there is only little overlap in the alterations of specific proteins between the HepG2 cells and our 16HBE14o^−^ cells, similar biological processes (e.g., oxidative stress, regulation of the cytoskeleton, and the extracellular matrix) seem to be affected by CYN exposure of the two cell types (Appendix A).

Taken together, exposure of airway epithelial cells to CYN may severely disturb several basal regulatory mechanisms in normal cell function, especially the attachment of these epithelial cells to the basal lamina, proper maturation of tight junctions, the rate of cell proliferation, and the propagation of mitosis from metaphase to anaphase and cytokinesis.

## 3. Materials and Methods

### 3.1. Chemicals and Reagents

Cylindrospermopsin was obtained from Enzo Life Sciences (Lörrach, Germany) and dissolved in 20% methanol in an aqueous solution (both HPLC grade). The stock (1 mmol/L) was aliquoted and stored at −20 °C. The rabbit polyclonal anti-CEP55 antibody was obtained from Life Technologies (Darmstadt, Germany). The rabbit polyclonal anti-SPARC antibody was obtained from Cell Signaling Technologies (Frankfurt/M., Germany). The mouse monoclonal anti-claudin-6 antibody was obtained from Santa Cruz Biotechnology (Heidelberg, Germany). For normalization of the Western blot signals, β-Actin-Ab #A5441 (obtained from Sigma Aldrich, St. Louis, MO, USA) was used at a dilution of 1:10,000. Horseradish Peroxidase-linked secondary antibodies (dilution of 1:6000) and enhanced luminescence reagents were obtained from Biozym, Oldendorf, Germany. An anti-rabbit Alexa Fluor 647-linked secondary antibody (111-605-003) was obtained from Jackson ImmunoResearch, Ely, United Kingdom, and used at a dilution of 1:500. All other chemicals were purchased from Carl Roth (Karlsruhe, Germany).

### 3.2. Experimental Cells

Experiments were performed using immortalized human airway epithelial cells (16HBE14o^−^) which had been derived from normal bronchial epithelial cells from a healthy donor [73,74]. Cells were cultured on 10 cm Cell^+^ dishes (Sarstedt, Numbrecht, Germany) in Eagle’s MEM (PAN-Biotech, Aidenbach, Germany) containing 10% FBS superior (Merck, Darmstadt, Germany), 29.8 mmol/L NaHCO_3_ and 1% (*w*/*v*) penicillin/streptomycin solution (PAN-Biotech, Aidenbach, Germany) at 37 °C and gassed with 5% CO_2_. The cell culture medium was changed every 3 days. Cells were exposed to CYN for different periods as soon as they had reached coverage of 80% of the cell culture plate surface. Cells had formed confluent monolayers (Figure 1) by the end of the experiments. All cell cultures were checked for *Mycoplasma* contaminations regularly.

### 3.3. Proteomic Analyses

Cells were cultured as previously described [75] and exposed to 5 µmol/L CYN or vehicle (0.1% methanol) for 20 h. Protein extracts containing 30 µg of protein were separated by 1D SDS-PAGE. The gel lanes were cut into 10 strips and the proteins in each strip were digested with trypsin at 37 °C overnight [76]. The resulting peptide mixtures were analyzed by an Ultimate 3000 (Thermo Scientific, Bremen, Germany) coupled witha QExactive Plus (Thermo Scientific, Bremen, Germany). After loading the peptides onto the analytical column (Acclaim PepMap RSLC C18, 2 µm, 100 Å, 75 µm ID × 25 cm) with buffer A (water in 0.1% formic acid), a binary 85 min gradient from 5% to 95% buffer B (80:20 acetonitrile in 0.1% formic acid: water in 0.1% formic acid) at a flow rate of 300 nL/min was used to separate the peptides in each mixture. Precursor spectra were acquired in profile mode in the scan range of 350 to 1600 *m*/*z* with a resolution of 70,000, an automatic gain control (AGC) of 1 × 10^6^, and a maximum injection time of 100 ms. A data-dependent acquisition (DDA) of the 10 most intense precursor ions was performed by collision-induced dissociation with a normalized collision energy of 27.5%, an AGC of 2 × 10^5^, and a maximum injection time of 100 ms. The isolation window was set to 2 *m*/*z*. Dynamic exclusion was set to 30 s and ions with unknown, single, or charges >7 were excluded from fragmentation.

The acquired raw data per condition were imported as fractions and processed together in the Proteome Discoverer (PD, version 2.5.0.400, Thermo Scientific, Bremen, Germany). The recalibration of the MS/MS spectra wasperformed via the implemented PD node “Spectrum Files RC” and by using the SwissProt database reduced to human entries (downloaded 06.07.2021, 23,542 entries) and the cRAP database (https://www.thegpm.org/crap/, downloaded 16 September 2020, 116 entries, accessed on 16 September 2020) with the following parameters: precursor tolerance of 20 ppm, fragment tolerance 0.5 Da, trypsin as a proteolytic enzyme, and carbamidomethylation of cysteine as fixed modification. For peptide-to-spectrum matching, the two-stage Sequest-HT search engine was performed using the above databases with the intensity-based INFERYS rescoring implemented in PD 2.5 [77,78,79]. In both phases of the database search, the following parameters were used: 0.02 Da fragment ion mass tolerance, 10 ppm parent ion tolerance, a maximum of two missed cleavages, trypsin as the digestion enzyme, default parameters for static and dynamic modifications, and strict false discovery rate (FDR) of 0.01 at the peptide and protein level by rescoring the spectra with Percolator [80,81]. Lable-free quantification was performed using the PD node “LFQ and precursor quantification” with the following parameters: unique peptides and razor peptides, precursor quantification via intensity, normalization via total peptide amount, calculation of protein abundance via summed abundances, protein ratios via pairwise peptide ratio calculation, and calculation of *p*-values via background-based *t*-test with Benjamini-Hochberg FDR correction. Protein identifications per biological replicate (n = 3) were filtered to accept only master proteins with at least two unique peptides and 5% sequence coverage.

Gene Ontology (GO)-terms of the UniProtKB-accessions were retrieved by the ID-mapping tool on the Uniprot webpage (https://www.uniprot.org/; accessed on 4 October 2022).

### 3.4. Western Blot Analysis

Cell culture, sample preparation, protein separation on SDS gels, and Western blotting were performed as previously described [75] on whole cell extracts of cells exposed to CYN (1, 2.5, or 5 µmol/L; selection of CYN-concentrations as used in previous studies [32,33,34,38]) or vehicle (0.1% methanol) for 0, 12, 24, or 36 h. Signals were recorded using an Intas ECL ChemoStar Imager (Intas, Göttingen, Germany). Band signal intensities were assessed by densitometry using Phoretix 1 D (Nonlinear Dynamics, Newcastle upon Tyne, UK). To correct for potential minor differences in exposure time, the mean density of all bands, or only control bands, on each gel image was used to normalize the densities of individual bands of the same gel. Signal intensities of the protein bands were normalized to the signals obtained in control samples only treated with 0.1% methanol (vehicle). Relative band densities were used to calculate the means and standard deviations of different experiments.

### 3.5. Fluorescence Microscopy

Cells were grown on coverslips until they reached 80% confluency. CYN (1, 2.5, or 5 µmol/L) or vehicle (0.1% methanol) were added for 0, 12, 24, or 36 h. The culture medium was aspirated, and cells were washed twice using phosphate-buffered saline (PBS) before they were fixed using 4% paraformaldehyde solution for 15 min at room temperature (RT). Cells were washed twice with PBS and permeabilized by exposing them to 0.1% (*w*/*v*) triton solution in PBS for 15 min at RT. Non-specific binding of antibodies was blocked by pre-exposing the cells to 2% (*w*/*v*) bovine serum albumin (BSA) in PBS for 60 min at RT. Incubation with primary antibodies (dilution 1:250 in blocking solution) occurred by the hanging drop method in a moist chamber overnight at 4 °C. Unbound primary antibody was washed away by washing the coverslips three times for 5 min in PBS at RT. The coverslips were incubated with the anti-rabbit Alexa Fluor 647-linked secondary antibody (dilution of 1:500) for 90 min at RT in the dark. Subsequently, the coverslips were washed three times with PBS for 5 min at RT. DNA staining was performed by incubation of the coverslips in 1 mg/L DAPI-solution in PBS for 10 min at RT. After washing the coverslips 5 times with PBS the fluid was allowed to run off. Mounting on microscopic slides occurred using a 1:1 mixture of Mowiol-488 and DABCO. The slides were dried overnight at 4 °C in the dark. Images of cells were taken using a Leica DMi8 fluorescence microscope with 10 × or 63 × lenses. Excitation and emission wavelengths were set to 358 or 461 nm for DAPI and 650 or 688 nm for Alexa Fluor 647, respectively. Image J was used to count the number of cells per image and the proportion of cells showing irregularly shaped nuclei or micronuclei. Example images were taken of cells in different phases of cell division, showing the presence of CEP55 at the spindles, or the spindles and the midbody, respectively, during ana- and telophases.

### 3.6. Data Presentation and Statistics

Data are presented as means ± S.D. of n experiments on different cell preparations. Significant differences in the series of means were detected by one-way ANOVA. Means were tested for significant differences to the appropriate controls using Student’s *t*-test if variances were equal (Lavene test), or otherwise, the Kolmogorov-Smirnov-test was used. Significant differences inmeans were presented as *p* < 0.05 (*), *p* < 0.01 (**), or *p* < 0.001 (***).

## Figures and Tables

**Figure 1 toxins-14-00785-f001:**
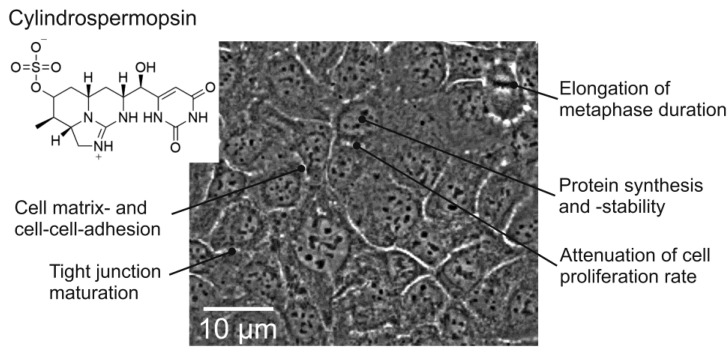
A frame of a time-lapse movie showing a portion of a confluent culture of 16HBE14o^−^ cells. Indicated are the major cell physiological mechanisms that were affected by cylindrospermopsin-treatment of cells according to the proteomic analyses. The chemical structure of cylindrospermopsin in the insert was created by Lukas Jacobsen using ChemDraw. The results of the proteome analysis were subjected to a search for functional clusters and the results are shown in the overrepresentation (Reactome) analysis scheme in Appendix A.

**Figure 2 toxins-14-00785-f002:**
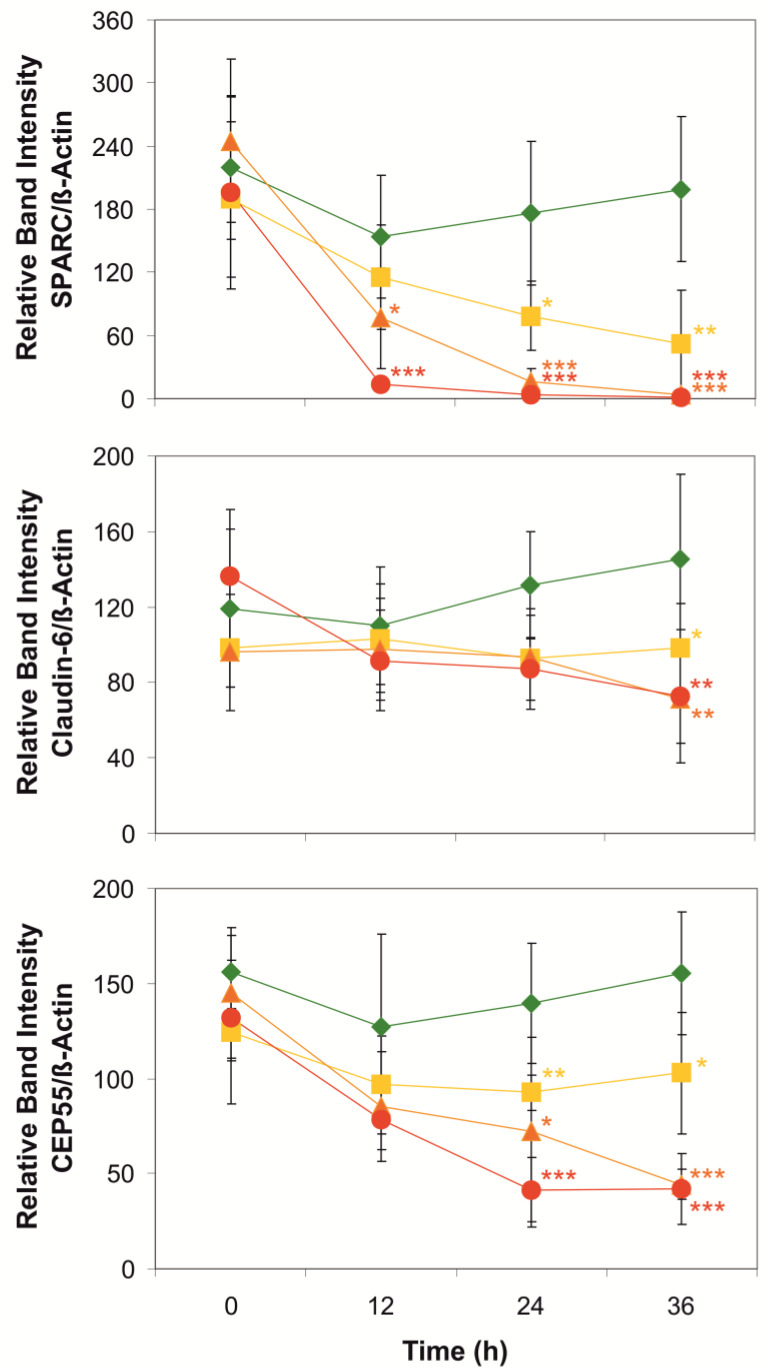
Western blot analyses of SPARC- (upper panel), claudin-6- (middle panel), and CEP55 expression in whole cell extracts of 16HBE14o^−^ cells upon exposure for 0, 12, 24, or 36 h to cylindrospermopsin (1 µmol/L—yellow squares; 2.5 µmol/L—orange triangles; 5 µmol/L—red dots) or vehicle (0.1% methanol—green diamonds). Data are presented as ratios of densities of antibody-labeled bands of the respective proteins and those of β-actin (internal standard). Means ± S.D., n = 4–10 biological replicates. Significant differences inmeans compared with those of the respective vehicle controls: *—*p* < 0.05; **—*p* < 0.01; ***—*p* < 0.001.

**Figure 3 toxins-14-00785-f003:**
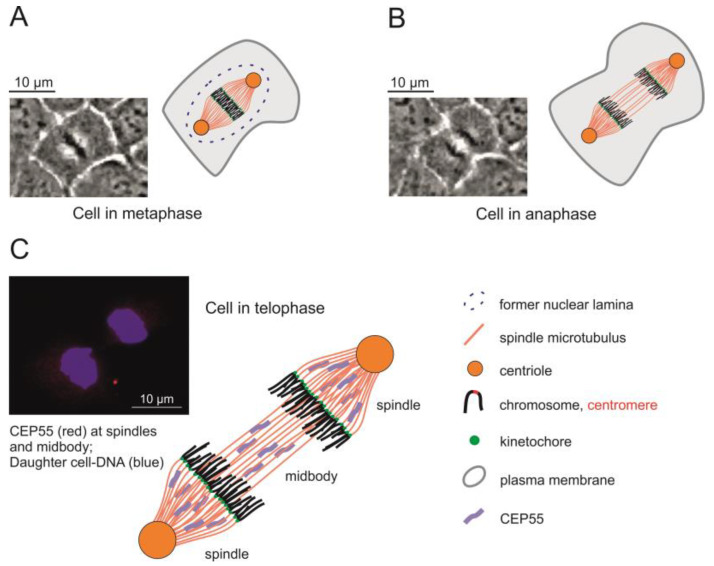
Schemes of 16HBE14o^−^ cells in metaphase (**A**) or anaphase (**B**) and visualization of CEP55 protein localization (**C**) at the spindle apparatus and the midbody (red fluorescence) relative to the DNA or still condensed chromosomes of the daughter cells (blue fluorescence), respectively, in late ana- and telophase of a mitotic cell. The microscopic images in (**A**,**B**) are stills taken from a time-lapse movie showing individual 16HBE14o^−^ cells in cell division. The microscopic image in (**C**) shows an overlay of fluorescent images of CEP55 labeled using an Alexa Fluor 647-linked antibody and of the daughter cell chromosomes stained with DAPI.

**Figure 4 toxins-14-00785-f004:**
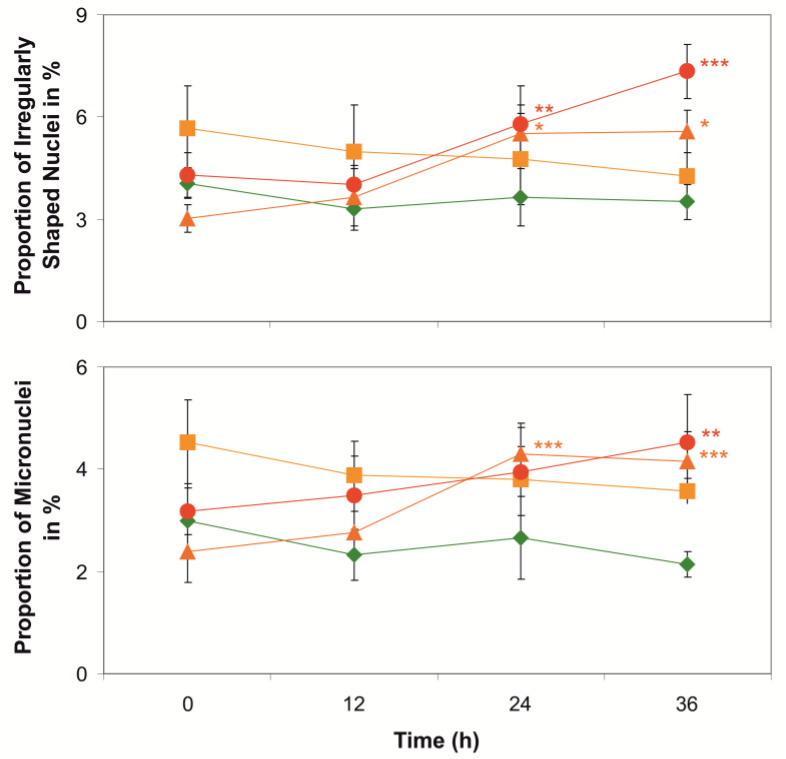
Analyses of the proportions of cells showing irregularly shaped nuclei (upper panel) or micronuclei (lower panel) in percent of all cells in cultures of DAPI-stained 16HBE14o^−^ cells upon exposure for 0, 12, 24, or 36 h to cylindrospermopsin (1 µmol/L—yellow squares; 2.5 µmol/L—orange triangles; 5 µmol/L—red dots) or vehicle (0.1% methanol—green diamonds). Means ± S.D., n = 4. Significant differences of means compared with the respective starting values at 0 h: *—*p* < 0.05; **—*p* < 0.01; ***—*p* < 0.001.

## Data Availability

All data included in this study are deposited within the submitted Appendix A.

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
