# Peer review of "Target Mechanisms of the Cyanotoxin Cylindrospermopsin in Immortalized Human Airway Epithelial Cells"

_toxins, 2022, doi:10.3390/toxins14110785_

Round 1

Reviewer 1 Report

The article titled "Target mechanisms of the cyanotoxin cylindrospermopsin in immortalized human airway epithelial cells” addresses about the effect that CYN has at airway level because of one of the possible routes of exposure to this cyanotoxin is the inhalation route. Authors have employed the 16HBE14o- cell line and applied proteomic studies.

This is an interesting article and provides new information on the toxicity of CYN, however, the following modifications need to be made.

Introduction

It is necessary to include more information about CYN, such as the producing species, mechanisms of action, toxicity, permitted levels of exposure in human, etc. It is necessary to add this information in the introduction. 

Line 29: Indicate due to what changes or what may lead to the occurrence of Harmful algal blooms (HABs).

Line 33-34: Indicate the main human exposure routes to cyanotoxins.

Line 39: Indicate the CYN-producing species

Line 42: … “zwitterionic character.”. Add a refence

Line 43-45: Mechanisms of action responsible for CYN toxicity are missing, review and complete.

Results and discussion

It would be interesting and enriching for the work that the authors compared the results obtained in this work with others obtained in different cell lines and that are also CYN target organs, in which the possible mechanisms and/or pathways are also evaluated through proteomic studies responsible for CYN toxicity.

Line 69: Justify the choice of the dose of CYN used in this work.

Line 78: Suppl. Fig.1 shows a synopsis of major cell physiological mechanisms that seemed to be affected by CYN-exposure in 16HBE14o- cells, it would be better if this figure is represented as separated figure due to the high information that contain.

Materials and methods

Explain the choice of the different doses of CYN used.

The authors have not indicated the conclusions of the work

Author Response

Introduction

It is necessary to include more information about CYN, such as the producing species, mechanisms of action, toxicity, permitted levels of exposure in human, etc. It is necessary to add this information in the introduction.

Author response: Additions have been made to the introduction about CYN-producing organisms, potential bioactivation of CYN by liver enzymes, and permitted levels of exposure in humans. We cited an addition publication on CYN toxicity (Pichardo et al. 2017) and supplemented previously raised hypotheses on how CYN affects eukaryotic cell systems.

Line 29: Indicate due to what changes or what may lead to the occurrence of Harmful algal blooms (HABs).

Author response: We added the information that HABs occur in water bodies that contain high levels of nutrients (nitrogen, phosphorus) and are exposed to intense sunlight.

Line 33-34: Indicate the main human exposure routes to cyanotoxins.

Author response: We added details about the potential uptake pathways of cyanotoxins in humans.

Line 39: Indicate the CYN-producing species

Author response: We have added infromatin on three major species of cyanobacteria producing CYN.

Line 42: … “zwitterionic character.”. Add a refence

Author response: Ref. Shaw, G. R., et al. (2000) has been added to the reference list.

Line 43-45: Mechanisms of action responsible for CYN toxicity are missing, review and complete.

Author response: Additions have been made to the introduction about CYN-producing organisms, potential bioactivation of CYN by liver enzymes, and permitted levels of exposure in humans. We cited an addition publication on CYN toxicity (Pichardo et al. 2017) and supplemented previously raised hypotheses on how CYN affects eukaryotic cell systems.

Results and discussion

It would be interesting and enriching for the work that the authors compared the results obtained in this work with others obtained in different cell lines and that are also CYN target organs, in which the possible mechanisms and/or pathways are also evaluated through proteomic studies responsible for CYN toxicity.

Author response: We added a paragraph on the comparison with another proteomic study of CYN effects in eukaryotic cells. HepG2 cells [Ref. Liebel, S., et al. (2016) showed certain coincidences in the biological processes that were affected by CYN exposure.

Line 69: Justify the choice of the dose of CYN used in this work.

Author response: This concentration is approximately 1/3 of the lethal dose in mice [Ref. Seawright et al. 1999] and results in less than 50% loss in cell viability over a period of 48 h [Ref. Kubickova et al. 2019b]. Concentrations of up to 2.5 μmol/l have been found to be non-toxic for human HepG2 cells [Ref. Liebel, S., et al. (2015). Other studies used CYN concentrations in the same range (1 – 5 μmol/l) [Refs. Lankoff et al. 2007; Straser et al. 2013; Zegura et al. 2004]. Furthermore, this study contributes to the chemical hazard characterisation of CYN exposure to the airways. Hazard characterisation needs to cover concentrations above those routinely monitored in the environment to also account for extreme exposure events (e.g., accidents). For this purpose, OECD guidelines generally recommend testing chemicals up to (cyto-) toxicity or solubility limits; coverage of concentrations up to 100 uM is not unusal, and in fact desired, in in vitro test methods for hazard characterisation.

Line 78: Suppl. Fig.1 shows a synopsis of major cell physiological mechanisms that seemed to be affected by CYN-exposure in 16HBE14o- cells, it would be better if this figure is represented as separated figure due to the high information that contain.

Author response: We have changed the figure configuration according to the reviewer’s suggestion. However, we left the Reactome Analysis extra as Suppl. Fig. 1 due to the fact that the entries could not be properly read if the image had been included into the manuscript body.

Materials and methods

Explain the choice of the different doses of CYN used.

Author response: We used CYN in the concentration range of 1 to 5 μmol/l. The justification of the highest concentration has been given above and has now been included in the manuscript. The other concentrations were used to obtain dose/effect-relations which indicate that the observed changes are biologically meaningful and actually due to exposure of cells to CYN.

The authors have not indicated the conclusions of the work

Author response: We made a clear statement of the main conclusions from this work in the ‘Key Contribution’ section: “Exposure of human airway epithelial cells to the cyanotoxin cylindrospermopsin resulted in downregulation of proteins belonging to several functional clusters, mainly regulation of protein stability, cellular adhesion and integration in the extracellular matrix, cell proliferation, cell cycle regulation, and completion of cytokinesis. The results of this proteomic study highlight these cell physiological mechanisms as potential toxin targets in eukaryotic cells in general.”

Reviewer 2 Report

The manuscript "Target mechanisms of the cyanotoxin cylindrospermopsin in 2 immortalized human airway epithelial cells" is very straightforward and well-written.

Although i have some minor suggestions/queries regarding the methodology used in the experiment.

1. On what basis  5 μmol/l of CYN was used for treatment?

2. It would be nice to include transcriptomics along side proteomics to have a clear picture on downregulation of key genes.

3. How about the stability of CYN during the treatment? As it has been stated that no immediate affect was observed so the treatment was done for 20h. 

4. Whats the mechanism by which CYN enters the cells or it just attached to membrane proteins and alters the function. Is it possible to separate membrane protein and cytosol to get more in depth picture of mechanism of action.

5. Line 194: Inspection of the 16HBE14o- cells using fluorescence microscopy revealed that CYN at the higher concentrations (2.5 and 5 μmol/l) and at the longer exposure times.... Its not higher concentration but 5 μmol/l was used for proteome analysis. 

6. Figure 2A and B should be enlarged for better view.

Author Response

The manuscript "Target mechanisms of the cyanotoxin cylindrospermopsin in 2 immortalized human airway epithelial cells" is very straightforward and well-written.

Although i have some minor suggestions/queries regarding the methodology used in the experiment.

On what basis 5 μmol/l of CYN was used for treatment?

Author response: This concentration is approximately 1/3 of the lethal dose in mice [Ref. Seawright et al. 1999] and results in less than 50% loss in cell viability over a period of 48 h [Ref. Kubickova et al. 2019b]. Concentrations of up to 2.5 μmol/l have been found to be non-toxic for human HepG2 cells [Ref. Liebel, S., et al. (2015). Other studies used CYN concentrations in the same range (1 – 5 μmol/l) [Refs. Lankoff et al. 2007; Straser et al. 2013; Zegura et al. 2004].

It would be nice to include transcriptomics along side proteomics to have a clear picture on downregulation of key genes.

Author response: We agree with the reviewer that this would be nice, but it exceeds our possibilities (time, money, personnel) to do this in a reasonable period of time.

How about the stability of CYN during the treatment? As it has been stated that no immediate affect was observed so the treatment was done for 20h.

Author response: CYN is a metabolically stable molecule [Ref. Chiswell, R. K., et al. (1999); citation has been added in the revised version of the MS) that has been shown to keep up its biological activity for at least 48 h in human airway cell culture [Ref. Kubickova et al. 2019b].

Whats the mechanism by which CYN enters the cells or it just attached to membrane proteins and alters the function. Is it possible to separate membrane protein and cytosol to get more in depth picture of mechanism of action.

Author response: This is a good question which has not been answered yet. Both pathways seem possible. Our protein extraction technique enriches intracellular proteins. However, some membrane proteins have also been identified in our extracts, but, generally, the analysis of such proteins is difficult by standard proteomic techniques.

Line 194: Inspection of the 16HBE14o- cells using fluorescence microscopy revealed that CYN at the higher concentrations (2.5 and 5 μmol/l) and at the longer exposure times.... Its not higher concentration but 5 μmol/l was used for proteome analysis.

Author response: This concentration was selected for reasons that are explained above (2). In addition, our reasoning was that we wanted to have a clear biological response as the repetitions of proteomic analyses have always a high degree of variability for technical reasons.

Figure 2A and B should be enlarged for better view.

Author response: This has been done.

Reviewer 3 Report

This paper shows a thorough and extensive piece of research, from which reasonable conclusions were drawn. It also demonstrrates the complexity of he system investigated.One point not discussed is the major difference between cyn response between cell types. Hepatocytes are killed quickly, probably due to cyn metabolism to a rapidly toxic compound through a liver specific oxidation route.

Author Response

This paper shows a thorough and extensive piece of research, from which reasonable conclusions were drawn. It also demonstrrates the complexity of he system investigated.One point not discussed is the major difference between cyn response between cell types. Hepatocytes are killed quickly, probably due to cyn metabolism to a rapidly toxic compound through a liver specific oxidation route.

Author response: We have added a paragraph on the potential bioactivation of CYN by CYP enzymes in liver cells. However, the presence of such a mechanisms that would explain a higher level of toxicity of CYN in liver cells [Ref. Norris et al. 2002] compared with other cell systems is controversial  [Refs. Kittler et al. 2016; Raska et al. 2019].

Round 2
